# The Impact of Training Time on Understanding the Treatment of Adolescent Idiopathic Scoliosis According to SOSORT International Guidelines: An Online Survey Study

**DOI:** 10.3390/children9111673

**Published:** 2022-10-31

**Authors:** Natália Albim Linhares, Juliene Corrêa Barbosa, Patrícia Jundi Penha, Rodrigo Mantelatto Andrade, Ana Paula Ribeiro, Mauricio Oliveira Magalhães

**Affiliations:** 1Master’s Program in Health, Environment and Society in the Amazon, Institute of Health Sciences (ICS), Federal University of Pará, Belém-Pará 66050-160, Brazil; 2Master’s Program in Human Movement Sciences, Institute of Health Sciences (ICS), Federal University of Pará, Belém-Pará 66050-160, Brazil; 3Department of Theories and Methods in Speech and Physical Therapy, Pontifical Catholic University of São Paulo, São Paulo 05015-901, Brazil; 4Physical Therapy Department, School of Medicine, University of São Paulo, São Paulo 01246-903, Brazil; 5Biomechanics and Musculoskeletal Rehabilitation Laboratory, Health Science Post-Graduate Department, Medicine School, University Santo Amaro, São Paulo 04829-300, Brazil

**Keywords:** scoliosis, physical therapy modalities, adolescent, knowledge, Brazil

## Abstract

The lack of knowledge by health professionals in the management of scoliosis can lead to an erroneous diagnosis. The aim of the current study was to understand the influence of length of professional experience on the knowledge of Brazilian physical therapists regarding international guidelines for the treatment of adolescent idiopathic scoliosis. The study was conducted using an online questionnaire consisting of 23 questions, divided into 8 questions on sociodemographic data and 15 questions based on information provided by the guidelines in the SOSORT 2016. In total, 506 physiotherapists were interviewed, and they comprised the study sample. The results showed that professionals who have been working for more than 6 years in the field have greater knowledge of AIS, seek to become more professional, and with that, have an understanding of AIS that is more aligned with the SOSORT. Length of experience made a difference when considering knowledge of the treatment of idiopathic scoliosis. The present study showed that a time since graduation of 6 years or more was satisfactory.

## 1. Introduction

Scoliosis refers to a three-dimensional deformity of the spine, causing lateral curvature in the frontal plane, axial rotation in the horizontal axis, and alteration of physiological curvatures in the sagittal plane [1]. Adolescent idiopathic scoliosis (AIS) is the most common spinal deformity, affecting 1 to 4% of adolescents, and progressing most rapidly during the pubertal growth spurt [1,2]. The prevalence in the general adolescent population is approximately 0.47–5.2% [3]. 

AIS is a diagnosis of exclusion, performed by anamnesis, physical examination, and imaging, determined in individuals with a Cobb angle greater than 10° and in the absence of neuromuscular disorders or congenital vertebral malformations [1]. Idiopathic scoliosis is of an unknown origin and is multifactorial. The higher risk of developing AIS is observed between ages 11–14 [1,4]. In addition, there are hypotheses related to genetic causes, alterations in growth potential, nutritional, biomechanical, and hormonal factors, and neurological abnormalities [2,5]. 

AIS can be treated according to the severity of the curves; severe curves with a Cobb angle above 50° are usually associated with surgery, aimed at reducing the deformity before the patient reaches skeletal maturity, while mild (10°–25°) and moderate curves (25°–45°) can be treated with conservative treatment, orthoses, and specific exercises for scoliosis [2,3]. A recent meta-analysis also suggested that exercise therapy has a potential effect on reducing the Cobb angle, trunk rotation angle, and improving quality of life in patients with adolescent idiopathic scoliosis [6]. Therefore, it is necessary for professionals to update themselves in the management of scoliosis according to the existing scientific evidence [1]. 

The Society for Orthopedic Treatment and Rehabilitation of Scoliosis (SOSORT) guidelines provide recommendations on standards for the management of idiopathic scoliosis, with the aim of ensuring a minimum quality of care, increasing the effectiveness of conservative treatment, and maximizing adherence to orthotic prescriptions [1]. The SOSORT was founded in 2004 to increase scientific knowledge, contribute to consumer education, and develop collaboration around better, non-operational management for all spinal deformities. 

The lack of knowledge of health professionals in the management of scoliosis can lead to erroneous diagnosis, treatment and follow-up of the condition. Scoliosis is a highly progressive pathology, which can lead to consequences such as: worsening of the pathological condition, limitation of patient’s functional capacity and biomechanics, exercise capacity, general fitness, and ability to work, all of which directly affect quality of life. However, not all AIS progress in the same way. The more severe the deformation the further the possibilities of progression. However, moderate AIS, especially in those with a late debut, is not scarily progressive [1,7]. 

The consequences that can occur due to the lack of knowledge of the appropriate treatment include prolongation of the treatment time and surgical indication arising from the worsening of the patient’s condition. Therefore, to efficiently treat the deformity, the diagnosis needs to be made as soon as possible and the treatment carried out by a multidisciplinary team with adequate knowledge [1,8]. Our hypothesis is that physical therapists with more than 6 years of training performed better, which may be related to more complete training, resulting from factors such as: a greater number of professional courses taken, presence in scientific congresses, and greater reading of professional articles, which contribute to the expansion of knowledge. However, for more studies, it is necessary to understand the impact of training time on understanding the treatment of adolescent idiopathic acoliosis according to the SOSORT international guidelines

Thus, the objective of the study was to understand the influence of time of professional experience on Brazilian physical therapists’ knowledge of the international guidelines (SOSORT) for the treatment of AIS.

## 2. Methods

This quantitative, observational, descriptive cross-sectional study included Brazilian physical therapists recruited by convenience, with previous experience in treating patients with adolescent idiopathic scoliosis (i.e., treated at least one patient in the previous six months). The survey was carried out online on the digital platform Google Forms, through the creation of an online form, which contained multiple-choice, discursive questions and evaluations on numerical scales. We are using a novel and non-validated questionnaire (Appendix A).

Physical therapists with regular registration in the regional council of physiotherapy and occupational therapy (CREFITO) were included in this study. Physical therapists who do not work with musculoskeletal rehabilitation, not having treated AIS patients, academics, or any other health professional were excluded. After the detailed explanation of the study, the physical therapist signed the informed consent form (ICF) before answering the questionnaire. This study was approved by the ethics committee of the Federal University of Pará (CEP: 4.134.796) with the collaboration of the Escoliosis Brasil—Institute, specialized in the conservative treatment of scoliosis (SP). 

## 3. Data Collection

The study population was invited for convenience to participate in the research, through social media platforms, including Instagram, Facebook, Twitter, and WhatsApp, on which reminders were placed fortnightly to promote voluntary participation in the study. Initially, questions were answered regarding sociodemographic data, such as name, age, gender, region in which they work (north, northeast, mid-west, southeast, and south), public or private undergraduate institution, training time, level of academic education (undergraduate, graduate, master’s, or doctorate), time of experience in caring for patients with idiopathic scoliosis, average number of patients seen in the previous 6 months, main place of work (private office or clinic, public service, home care, gym, or hospital), and main technique used in the care of these patients. 

The participants then answered a questionnaire previously prepared by the researchers, based on information provided by the SOSORT 2016 guidelines. The survey was divided into distinct sections with questions linked to the analysis of scoliosis knowledge (definition, cause, development, diagnosis, conservative treatment, and indication of orthopedic brace) and two multiple-choice questions, to analyze the practitioners’ opinions on treatment indication and familiarity with the method they thought would be beneficial for the patient’s scoliosis, as well as the physical activity indication for patients with idiopathic scoliosis. Initially, this questionnaire was reviewed and approved by physical therapists specialized in the treatment of AIS.

## 4. Statistical Analysis 

Data were analyzed using descriptive statistics, calculation of percentages, and a survey of frequency of cases. The output results are presented in graphic representation or in the form of a table.

## 5. Results

### Participants

Of the 506 physical therapists interviewed who participated in the study, 79.6% were female (*n* = 403), with a mean age of 33.5 ± 8.1 years, and 20.3% were male (*n* = 103), with a mean age of 34.0 ± 8.0 years. The physical therapists interviewed held undergraduate (bachelor’s), postgraduate, master’s, doctoral, and postdoctoral degrees, as shown in Figure 1.

Table 1 describes the demographic and professional characteristics of the physical therapists. The majority of the professionals work in the private sector and in offices. In addition, they treated 1–5 patients regardless of the time since graduation and did not have a specific technique for the treatment of adolescents with idiopathic scoliosis. Survey participants were grouped according to time of experience into the following ranges: less than 1 year, 1 to 3 years, 4 to 6 years, and over 6 years.

When considering the knowledge of scoliosis among Brazilian physical therapists working in groups with different lengths of experience, it can be observed that most professionals have little knowledge about the cause and screening tests for AIS (Table 2).

Figure 2 describes in percentage numbers the most recommended conservative method for the treatment of idiopathic scoliosis. The result showed that the indication of specific exercises is more prevalent in the groups between 4–6 years and >6 years of professional experience. However, the junior professional is the one who specializes according to the SEAS (Scientific exercise methods Approach to Scoliosis) approach to scoliosis treatment (<1 year of experience). Furthermore, 393 physical therapists indicate specific exercises for scoliosis as the best method.

Figure 3 describes, in percentage numbers, the physical activities that benefit patients with scoliosis and the personal choice of each physical therapist regarding the activity they consider to be the most recommended. The results showed that Brazilian physical therapists, in general between the experience years, recommend the practice of any of the physical activities (325) shown, among them: yoga, swimming, Pilates.

## 6. Discussion

The present study sought representatives from all regions of Brazil and included 506 Brazilian physical therapists. The study aimed to understand the influence of time of professional experience on the knowledge of Brazilian physical therapists regarding international guidelines in the treatment of adolescent idiopathic scoliosis. The results showed that, regardless of the years of experience, the items with greater professional knowledge aligned with the SOSORT, were related to when AIS develops In addition, a trained Brazilian physical therapist completes courses with knowledge of specialized techniques in the treatment of scoliosis. However, experienced professionals (trained physiotherapists with 4–6 years or more experience) seek knowledge in paid short courses and conferences. This update is important in the treatment of patients with idiopathic scoliosis.

In addition, the results showed physical therapy exercises are recommended to pre-vent the progression of AIS, and, with that, the professionals with an understanding of adolescent idiopathic scoliosis are more aligned with the SOSORT. Therefrom, we were able to conclude that professionals recognize the best conduct to be performed in these patients, such as treatment through specific exercises for scoliosis. However, as the specializations in specific exercises were created in Europe, the improvement in Brazilian physical therapist professionals can be affected due to the difficulty of accessing these specializations. 

In addition, considering the questions about the causes of AIS, the majority of professionals, of all ranges of years of experience, answered incorrectly, demonstrating their lack of knowledge about the problem. Furthermore, Brazilian physical therapists showed greater knowledge of the Risser classification, a more specific analysis, when compared to the Adams test, which is a simpler screening method. 

Given this, it seems that there is a failure in primary learning, which needs to be corrected, in order to improve the efficiency of the AIS identification process. This indicates that Brazilian physical therapists are poorly equipped to provide a satisfactory level of care to scoliosis patients in daily clinical practice. Recently, a study carried out including 165 physiotherapy students from the UK, with the aim of analyzing the current knowledge on understanding idiopathic scoliosis, referring to the 2011 SOSORT guidelines, concluded that their knowledge is insufficient regarding the management of idiopathic scoliosis [8]. This may lead to a large number of patients without a correct diagnosis or receiving a delayed diagnosis and, thus, may contribute to worsening of the prognosis, increasing the risk of surgical intervention, and physical and emotional dysfunctions. 

Thus, when compared to the current study, we can observe that there are still flaws in the learning process, in the primary methods of screening and evaluation, which can directly influence the correct diagnosis and therapeutic conduct [8]. Therefore, it is important that Brazilian physical therapists seek continuous training, through reading scientific articles, participating in scientific events, conducting courses on scoliosis assessment and treatment and specializations in the area in order to apply more efficient treatment approaches to patients. Another survey of 37 physical therapy students in Poland showed that physical therapists are taught according to SOSORT guidelines, so they are much more familiar with the etiology of scoliosis, along with the treatment approaches available to this group of patients [9].

Furthermore, a recent study [10] sought to investigate the knowledge of Brazilian physical therapists regarding the management of low back pain. The authors concluded that there is also a failure in the primary learning process, resulting, sometimes, in an incorrect diagnosis, which, in turn, delays the patient’s treatment, and may lead to incorrectly performed treatment [11]. The data indicate that physical therapists should be better trained to identify serious conditions that require referral to specialist services. These results are similar to those found in the current study, which once again reinforce the importance of knowing the tests used in the primary screenings of musculoskeletal disorders, aiming at a correct diagnosis and, later, more efficient treatment.

Furthermore, a recent study [12] sought to determine the awareness of physical therapists about physical activity and exercise prescription. The study included 1.352 physical therapists from 56 different countries and reported the result that 60% of respondents correctly stated the physical activity guidelines for adults and children, however the majority of respondents (79%) believe the lack of opportunity for professional development has affected their ability to prescribe exercise, concluding that many physical therapists lack the knowledge and training to provide advice on physical activity and prescribe aerobic exercise and resistance training for people with musculoskeletal pain.

In addition, some factors can influence the conduct and effectiveness of the physical therapists, among which we can highlight the difference between genders and the time since graduation. A study published in 2014, which analyzed the views of men and women on low back pain, identified that women have an emotional focus on the individual, while men have a physical focus, In this way, these different views directly influence the individual way in which the professional will carry out their duties [13]. 

Furthermore, a study concluded that Brazilian physical therapists do not use the best available evidence for clinical decision-making, therefore, they do not apply practices that are scientifically more effective in patient treatment [10], which contradicts what is written in the physical therapy guidelines, which state that the physical therapist must be qualified and up-to-date to care for patients with different pathologies [14]. The SOSORT guidelines are free and are the easiest and most reliable way for the clinical physical therapist to use the best scientific evidence in the conservative treatment of scoliosis, therefore, their use should be further explored.

In the present study, physical therapists with more than 6 years of training performed better, which may be related to more complete training, resulting from factors such as: a greater number of professional courses taken, presence in scientific congresses, and greater reading of professional articles, which contribute to the expansion of knowledge. In addition, the study showed that professionals with less than 6 years of training do not have an understanding of the best practices to be performed in patients with adolescent idiopathic scoliosis (AIS) [3,15,16,17,18,19]. 

A possible limitation of the study is that there was no sample size calculation to define the exact number of physical therapists needed to form the sample. However, there is no estimate of physical therapists who work in the trauma–orthopedic area and more specifically who treat patients with scoliosis in Brazil. Furthermore, the results are based on a non-validated questionnaire, and we propose this future validation as a new contribution of their research.

## 7. Conclusions

The present study sought representatives from all regions of Brazil and included 506 Brazilian physical therapists. The study aimed to understand the influence of time of professional experience on the knowledge of Brazilian physical therapists regarding international guidelines in the treatment of AIS. The results showed that, regardless of the years of experience, the items with greater professional knowledge according to the SEAS aligned with the SOSORT were related to when AIS develops and the recommended physical therapy exercises to prevent the progression of AIS. In addition, we were able to conclude that professionals recognize the best practices to be performed in these patients, such as treatment through specific exercises for scoliosis, however, as the specializations in specific exercises were created in Europe, the improvement in Brazilian physical therapist professionals can be affected due to the difficulty of accessing these specializations.

## Figures and Tables

**Figure 1 children-09-01673-f001:**
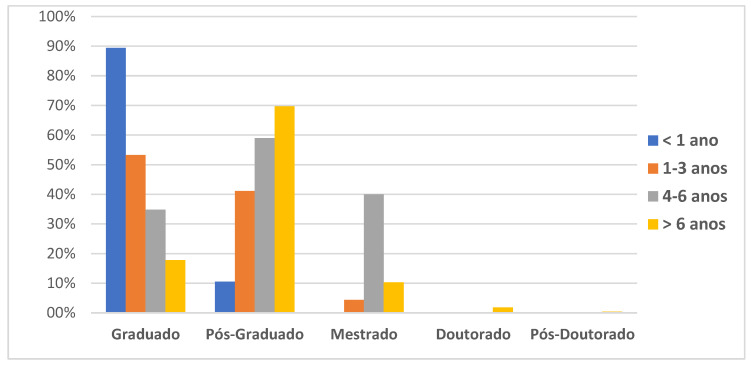
Representation of the participants’ academic background.

**Figure 2 children-09-01673-f002:**
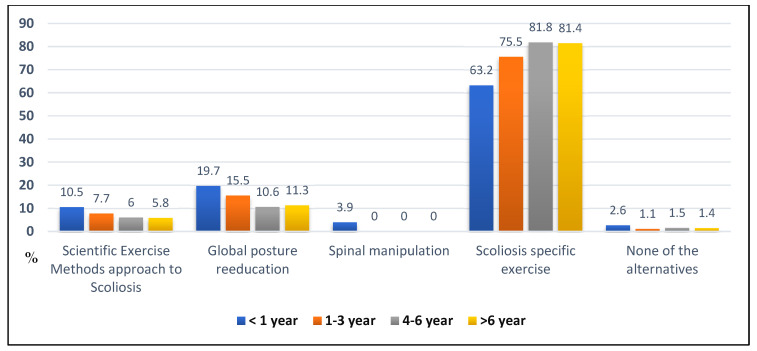
Percentual of participants opinion on the conservative method that would be recommended for the treatment of AIS.

**Figure 3 children-09-01673-f003:**
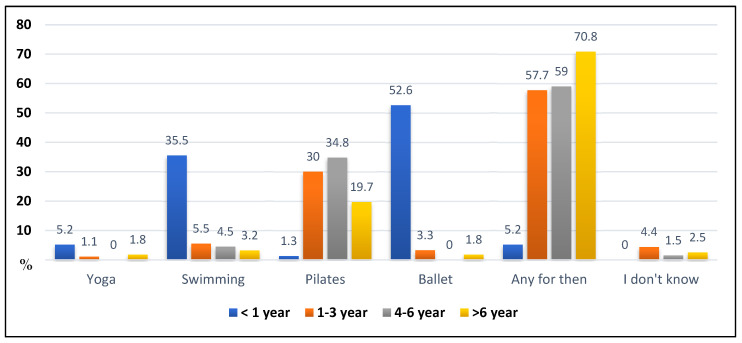
Opinions of the participants on physical activities that benefit patients with AIS.

**Table 1 children-09-01673-t001:** Demographic and clinical data of the 506 Brazilian physical therapists, presented according to time since graduation.

Experience Time	Public	Private		
	Yes (%)	Yes (%)		
Less than 1 year	13 (17.1)	63 (82.8)		
Between 1 and 3 years	23 (25.5)	67 (74.4)		
Between 4 and 6 years	18 (27.3)	48 (72.7)		
More than 6 years	40 (14.6)	234 (84.4)		
Operating region	<1 year (*n* = 76)	1–3 years (*n* = 90)	4–6 years (*n* = 66)	>6 years (*n* = 274)
	Yes (%)	Yes (%)	Yes (%)	Yes (%)
North	13 (17.1)	08 (8.9)	07 (10.6)	11 (4.7)
Northeast	31 (40.7)	31 (34.4)	24 (36.4)	63 (27.0)
Midwest	09 (11.8)	09 (10.0)	05 (7.6)	22 (9.4)
Southeast	19 (25.0)	89 (98.8)	17 (25.7)	149 (63.6)
South	04 (5.3)	07 (7.8)	13 (19.7)	29 (12.4)
Place of Work	<1 year (*n* = 76)	1–3 years (*n* = 90)	4–6 years (*n* = 66)	>6 years (*n* = 274)
	Yes (%)	Yes (%)	Yes (%)	Yes (%)
Clinic	25 (32.9)	45 (50.0)	48 (72.7)	273 (99.6)
Public servant	2 (2.6)	3 (3.3)	13 (19.6)	30 (10.9)
Home care	32 (0.0)	24 (26.6)	14 (21.2)	16 (5.8)
Gym	0 (0.0)	0 (0.0)	01 (1.5)	6 (2.2)
Hospital	02 (2.6)	01 (1.1)	04 (6.0)	0 (0.0)
Other	15 (19.7)	07 (7.7)	0 (0.0)	19 (6.7)
Number of patients seen	<1 year (*n* = 76)	1–3 years (*n* = 90)	4–6 years (*n* = 66)	>6 years (*n* = 274)
	Yes (%)	Yes (%)	Yes (%)	Yes (%)
1 to 5	74 (97.3)	84 (93.3)	55 (83.3)	207 (88.5)
5 to 10	2 (2.7)	6 (6.6)	6 (9.0)	40 (17.0)
>10	0 (0.0)	0 (0.0)	5 (7.6)	24 (10.2)
Specific treatment technique	<1 year (*n* = 76)	1–3 years (*n* = 90)	4–6 years (*n* = 66)	>6 years (*n* = 274)
	*n* (%)	*n* (%)	*n* (%)	*n* (%)
Yes	23 (30.2)	31 (34.4)	45 (68.2)	194 (83.0)
No	53 (69.8)	59 (65.6)	21 (31.8)	80 (34.2)
Technical training	<1 year (*n* = 76)	1–3 years (*n* = 90)	4–6 years (*n* = 66)	>6 years (*n* = 274)
	Yes (%)	Yes (%)	Yes (%)	Yes (%)
Presential course	16 (21.0)	30 (33.3)	35 (53.0)	164 (59.8)
Online course	6 (7.8)	4 (4.4)	8 (12.1)	32 (11.6)
Symposium	1 (1.3)	5 (5.5)	2 (3.0)	9 (3.2)
Not done	53 (69.7)	51 (56.6)	21 (31.8)	69 (25.8)

Data presented in absolute numbers (%).

**Table 2 children-09-01673-t002:** Number of correct answers on knowledge about SOSORT according to the time since training of Brazilian physiotherapists.

	Correct Answers *n* (%)			
Questions	<1 year (*n* = 76)	1–3 years (*n* = 90)	4–6 years (*n* = 66)	>6 years (*n* = 274)
What is AIS *?	72 (94.74) **	86 (95.5) **	61 (92.4)	243 (88.6)
What are the causes of AIS?	0 (0.0)	5 (5.5)	1 (1.5)	8 (2.9)
When does AIS develop?	72 (94.7) **	86 (95.5) **	61 (92.4) **	263 (95.9) **
What is the prevalence of AIS among scoliosis in general?	11 (14.4)	7 (7.7)	8 (12.1)	27 (9.8)
When is the diagnosis confirmed?	57 (75)	60 (66.6)	35 (53.0)	167 (61.3)
What does Risser represent?	54 (71.0)	47 (52.2)	40 (60.6)	180 (65.6)
What is the Adams test?	7 (9.2)	11 (12.2)	5 (7.5)	40 (14.5)
How is the referral to the physical therapist made?	40 (52.6)	54 (60.0)	33 (50.0)	153 (55.8)
When is conservative treatment recommended?	53 (69.7)	60 (66.6)	27 (40.9)	149 (54.3)
What are the goals of conservative treatment?	70 (92.1) **	77 (85.5)	61 (92.4) **	241 (87.9)
When is the use of a brace recommended?	55 (72.3)	50 (55.5)	41 (62.1)	169 (61.6)
What are the recommended physical therapy exercises to prevent the progression of AIS?	76 (100) **	86 (95.5) **	60 (90.9) **	256 (93.4) **
What are the recommendations for patients with AIS who practice physical activity?	62 (81.5)	78 (86.6)	57 (86.3)	236 (86.1)

Legend: * AIS (adolescent idiopathic scoliosis); ** Higher percentage of knowledge of SEAS.

## Data Availability

Data are available upon request addressed to the contact author.

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
