# Peer review of "The Impact of Training Time on Understanding the Treatment of Adolescent Idiopathic Scoliosis According to SOSORT International Guidelines: An Online Survey Study"

_children, 2022, doi:10.3390/children9111673_

Round 1

Reviewer 1 Report

Dear editor, please find in this document my review considerations for the authors:

INTRODUCTION

-          Line 42-43: If you are referring to adolescent idiopathic scoliosis, age is between 10-18. If you are referring to age at higher risk of developing AIS, it is between 11-14. And if you are talking about idiopathic scoliosis itself, it could emerge even before the age of 10 (EOS).

-          Line 48: degree sign is needed here and in all the rest of the text

-          Line 64: Not only erroneous diagnosis but also treatment and follow-up of the condition (suggestion)

-          Line 64: I recommend you to point out that not all AIS progress in the same way. The more severe the deformation the more the possibilities of progression. But moderate AIS, especially those with a late debut, are not scarily progressive

-          Line 65: Has this fact been proved? To my knowledge doing inadequate treatment is the same as doing nothing in AIS. And doing nothing does not worsen the condition, just let it progress according to its natural progression

-          Line 74-76: I miss here a paragraph that justify why do you think that years of professional experience may influence knowledge on SOSORT guidelines. A non-expert PT could have greater expertise in AIS consensus documents than a higher experienced one that does not made efforts in updating him/herself. Then, after reading all the paper and paying great attention at the results, less experienced PT´s showed, at least, similar correct answers regarding SOSORT rules than >6 years group (see table 2)

METHODS

-          Line 83: I don´t know if this link is necessary...

-          Line 104-111: The questionnaire used in this research should be included, at least as supplementary material for reader´s check. Also, a questionnaire always needed to be validated in different populations and by other researchers to be sure about its validity to answer the research question. Thus, I recommend the authors to:

o   Point out in the paper that they are using a novel and non-validated questionnaire

o   Share the questionnaire with the community so we can replicate it and help in the validation process

o   Express that results are based on a non-validated questionnaire, and propose this future validation as a new contribution of their research

RESULTS

-          Line 125: we use to call it figures, not graphs (please check the rest of figures of the paper)

-          In table 1: do percentages refer to experience groups or operating region groups? They don´t make sense to me in any of these two options anyway... because percentage sum is not 100 in any direction (rows or columns). The same apply to item “Place of work”

-          In table 1: this result must be discussed further because less experienced PT do not treat AIS significantly (more than 10 patients in the last 6 months), and if you want to measure the knowledge of SOSORT rules probably not having treated AIS patients should be a exclusion criteria…

-          In figure 2: percentages? it seems to be absolute numbers... Then, foot legend of the figure does not match with the names in x axis in the figure itself. Finally, I´m not an expert but quality of the figure (dpi) seems to be very low

-          In figure 3: again, there is a problem with % (they are absolute numbers according to the text). Also, as we cannot see the questionnaire, we don´t know if there were more sport options further than the ones reflected in the figure

DISCUSSION

-          Line 167: In which result do you base this conclusion? If you consider percentage of correct answers in table 2, all groups of years of experience show high % of correct responses to SOSORT questions. Differences between >6 and the rest of the groups were in number of patients seen, specific treatment use and technical training but not in correct answers of SOSORT questions in the questionnaire:

-          Line 181-184: Reference for this paper?

-          Line 202: I completely agree with this fact, to standardize clinical decisions we may enhance our protocols. This is well explained in Peterson, S., Shepherd, M., Farrell, J., & Rhon, D. I. (2022). The Blind Men, the Elephant, and the Continuing Education Course: Why Higher Standards Are Needed in Physical Therapist Professional Development. The Journal of orthopaedic and sports physical therapy, 1–14. Advance online publication. https://doi.org/10.2519/jospt.2022.11377

-          Line 233-234: This is, from my point of view, a great discordance between objective, methods, results and discussion. If your objective was to measure the influence of experience in the knowledge of SOSORT guidelines, your result (table 2) was that all groups showed higher % of correct answers about SOSORT questions. And you are giving a different conclusion, where >6 showed greater knowledge. The only result in which veteran PT´S showed better performance was number of patients seen, specific treatment use and technical training, items that are not aligned with your objective (Knowledge of SOSORT guidelines)

Author Response

Please find resubmitted our revised version of our article entitled “The impact of training time on understanding the treatment of Adolescent Idiopathic Scoliosis according to SOSORT international guidelines: an online survey study” (Manuscript ID - children-1915174). We appreciate the comments from the review team and from you. The manuscript has been revised to address the comments received from the reviewers. We addressed all points raised by the reviewers. We highlighted the new changes in underlined. We have provided a point by point response to each of the reviewer’s comments. We feel that these mayor changes have strengthened the manuscript and we would thank the reviewers for their helpful comments. I am looking forward to hear from you soon, Best wishes, Maurício Magalhães (on behalf of my co-authors)

Introduction

Point 1: If you are referring to adolescent idiopathic scoliosis, age is between 10-18. If you are referring to age at higher risk of developing AIS, it is between 11-14. And if you are talking about idiopathic scoliosis itself, it could emerge even before the age of 10 (EOS).

Response 1: We strongly appreciate and thank reporting of the revisor. We added this information in the manuscript. See lines 43-44

Point 2: degree sign is needed here and in all the rest of the text

Response 2: We agree with your comment. All the manuscript was revised. See lines 48-50

Point 3: Not only erroneous diagnosis but also treatment and follow-up of the condition (suggestion)

Response 3: We agree with your comment. See line 64

Point 4: I recommend you to point out that not all AIS progress in the same way. The more severe the deformation the more the possibilities of progression. But moderate AIS, especially those with a late debut, are not scarily progressive

Response 4: We have better explained the reasons AIS not progress in the same way, see lines 68-70.

Point 5: Has this fact been proved? To my knowledge doing inadequate treatment is the same as doing nothing in AIS. And doing nothing does not worsen the condition, just let it progress according to its natural progression 

Response 5: We agree with your comment. The manuscript was modified. See line 65-67

Point 6: I miss here a paragraph that justify why do you think that years of professional experience may influence knowledge on SOSORT guidelines. A non-expert PT could have greater expertise in AIS consensus documents than a higher experienced one that does not made efforts in updating him/herself. Then, after reading all the paper and paying great attention at the results, less experienced PT´s showed, at least, similar correct answers regarding SOSORT rules than >6 years group (see table 2)

Response 6: Thank you for your suggestion, we have modified the manuscript as suggested. See lines 75-81; 177-181

Point 7: Line 83: I don´t know if this link is necessary.

Response 7: Thank you for your suggestion, we have excluded this information in the manuscript. See line 90.

Point 8: The questionnaire used in this research should be included, at least as supplementary material for reader´s check. Also, a questionnaire always needed to be validated in different populations and by other researchers to be sure about its validity to answer the research question.

Response 8: We agree with your comment. We included the questionnaire in the supplementary material and modified the manuscript. See line 92 and 255-257.

RESULTS

Point 9. Line 125: we use to call it figures, not graphs (please check the rest of figures of the paper)

Response 9: We strongly appreciate and thank reporting of the revisor. We have modified in the manuscript. See lines 153-168).

Point 10. In table 1: do percentages refer to experience groups or operating region groups? They don´t make sense to me in any of these two options anyway... because percentage sum is not 100 in any direction (rows or columns). The same apply to item “Place of work”

Response: We strongly appreciate and thank reporting of the revisor. We modified the table 1 for better for better understanding. We hope to have answered your request. See line 144 (table 1).

Point 11. In table 1: this result must be discussed further because less experienced PT do not treat AIS significantly (more than 10 patients in the last 6 months), and if you want to measure the knowledge of SOSORT rules probably not having treated AIS patients should be a exclusion criteria…

Response 11: We strongly appreciate and thank reporting of the revisor. We modified the manuscript. See lines 96; 175-181.

Point 12. In figure 2: percentages? it seems to be absolute numbers... Then, foot legend of the figure does not match with the names in x axis in the figure itself. Finally, I´m not an expert but quality of the figure (dpi) seems to be very low

Response 12: We strongly appreciate and thank reporting of the revisor. We have replaced graph 2 with percentage data according to physical therapist experience time groups for better understanding. We hope to have answered your request. See lines 150-152; 160-162).

Point 13. In figure 3: again, there is a problem with % (they are absolute numbers according to the text). Also, as we cannot see the questionnaire, we don´t know if there were more sport options further than the ones reflected in the figure.

Response 13: We strongly appreciate and thank reporting of the revisor. We have replaced graph 3 with percentage data according to physical therapist experience time groups for better understanding. We hope to have answered your request. See lines 163-168; 169-170).

DISCUSSION

Point 14: Line 167. In which result do you base this conclusion? If you consider percentage of correct answers in table 2, all groups of years of experience show high % of correct responses to SOSORT questions. Differences between >6 and the rest of the groups were in number of patients seen, specific treatment use and technical training but not in correct answers of SOSORT questions in the questionnaire:

Response 14: We strongly appreciate and thank reporting of the revisor. The manuscript was modified. See line 175-181.

Point 15:  Reference for this paper?

Response 15: Thank you for this reference. The manuscript was adjusted. See line 201.

Point 16. Line 202: I completely agree with this fact, to standardize clinical decisions we may enhance our protocols. This is well explained in Peterson, S., Shepherd, M., Farrell, J., & Rhon, D. I. (2022). The Blind Men, the Elephant, and the Continuing Education Course: Why Higher Standards Are Needed in Physical Therapist Professional Development. The Journal of orthopaedic and sports physical therapy, 1–14. Advance online publication. https://doi.org/10.2519/jospt.2022.11377

Response 16: Thank you for this reference. See line 218

Point 17: This is, from my point of view, a great discordance between objective, methods, results and discussion. If your objective was to measure the influence of experience in the knowledge of SOSORT guidelines, your result (table 2) was that all groups showed higher % of correct answers about SOSORT questions. And you are giving a different conclusion, where >6 showed greater knowledge. The only result in which veteran PT´S showed better performance was number of patients seen, specific treatment use and technical training, items that are not aligned with your objective (Knowledge of SOSORT guidelines).

Reviewer 2 Report

This is interesting submission.

There is balance in all sections and adequate literature support.

Author Response

Dear Revisor.

Thank you for appreciating our work.

Please find attached the revised version

Best Regards

Mauricio Magalhaes and co authors.

Reviewer 3 Report

Dear Authors,

This is a well-designed and very useful study. Many aspects of the topic had been exposed in an appropriate way. Statistic strategy fits well the purposes of the study. Literature knowledge is good. The conclusion is in line with the results shown in the paper. Language is good for a scientific paper.

Study design: quantitative, observational, descriptive cross sectional study.

Aim: the objective of the study was to understand the influence of time in professional experience on Brazilian physical therapists’ knowledge of the international guide lines (SOSORT) for the treatment of AIS.

Population: Brazilian physical therapists

Collection: questionnaire for physical therapist in Brazil

Outcomes: not completely understandable. How did you grade the

Materials and Methods

The population target, the intervention, the collection has been clearly and adequately exposed. Could have been exposed better

Statistic: Statistical aspects are defined and explained well in the methods sections. The statistic is in line with the research question.

Results

The results has been clearly exposed.

Discussion

 The literature is well organized and the choice of the papers is large enough.

Conclusion

Conclusion is in line with the findings shown in the results.

Author Response

We really appreciate your comments about our work.

Please find attached the revision version of the manuscript.

Unfortunately, the phrase of your comment didn`t appear entirely to us. However, I hope that the revised version accomplished it.

Best Regards,

Mauricio Magalhaes and coauthors.

Round 2

Reviewer 1 Report

Please consider this mail as an acceptance for publication with my proffesional criteria and give the authors my personal recognition for their great work.   Kindest regards